# Goal-Driven Explainable Clustering via Language Descriptions

**Zihan Wang**
ziw224@ucsd.edu

**Jingbo Shang**
jshang@ucsd.edu

**Ruiqi Zhong**
ruiqi-zhong@berkeley.edu

## Abstract

Unsupervised clustering is widely used to explore large corpora, but existing formulations neither consider the users' goals nor explain clusters' meanings. We propose a new task formulation, "**Goal**-Driven Clustering with **Ex**planations" (GOALEX), which represents both the goal and the explanations as free-form language descriptions. For example, to categorize the errors made by a summarization system, the input to GOALEX is a corpus of annotator-written comments for system-generated summaries and a goal "*cluster the comments based on why the annotators think the summary is imperfect.*"; the outputs are text clusters each with an explanation ("*this cluster mentions that the summary misses important context information.*"), which relates to the goal and accurately explains which comments should (not) belong to a cluster. To tackle GOALEX, we prompt a language model with "[corpus subset] + [goal] + *Brainstorm a list of explanations each representing a cluster.*"; then we classify whether each sample belongs to a cluster based on its explanation; finally, we use integer linear programming to select a subset of candidate clusters to cover most samples while minimizing overlaps. Under both automatic and human evaluation on corpora with or without labels, our method produces more accurate and goal-related explanations than prior methods.

## 1 Introduction

Text clustering is widely used to explore large corpora (Aggarwal and Zhai, 2012). However, existing formulations cannot adapt to different users' goals, which might be clustering based on sentiment, genre, or other properties (Aharoni and Goldberg, 2020a); as a result, the desired output is underspecified. Furthermore, since the output clusters are not immediately interpretable, users must manually examine the clusters to gain insights. This can be time-consuming, especially when some clusters are semantically incoherent (Chang et al., 2009).

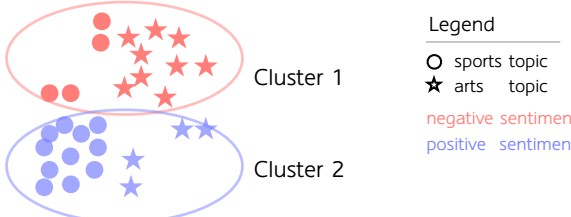

✔ **User's Goal** "*I want to cluster based on **sentiment***"

✔ **Explanation** "*Cluster 1 has a negative sentiment and Cluster 2 has a positive sentiment*"

Figure 1: An illustration of our task formulation GOALEX (**Goal**-Driven Clustering with **Ex**planations), where the input is a set of texts (corpus) and a goal, and the output constitutes a set of corpus subsets (clusters) each with an explanation. Given the goal, a successful GOALEX algorithm should cluster based on sentiment instead of topic and for each cluster explain which samples should (not) belong to it.

To address these weaknesses, we propose a new task formulation, GOALEX, "**Goal**-Driven Clustering with **Ex**planations" (Section 2). As illustrated in Figure 1, the input to the task is a text corpus with multiple attributes (e.g., sports and arts related texts with different sentiments) and a goal description in natural language ("*clustering based on sentiment*"). The output of the task constitutes a set of corpus subsets (clusters), each with a natural language explanation of which text samples should or should not belong to the cluster (e.g. "*contains positive sentiment*"). The output should satisfy three desiderata: 1) the explanations are goal-related, 2) each cluster is accurately described by its explanation, and 3) the clusters should overlap minimally while their union should cover most of the corpus.

To tackle GOALEX, we develop a three-stage algorithm **P**ropose-**A**ssign-**S**elect (PAS, Figure 2, Section 3), each designed to address one of the desiderata. At the proposal stage we address the 1st desideratum that the explanation should be goal-related; we do this by prompting a language model (LM) to generate a list of goal-related explanations

**❶ Propose**: given a goal and a set of texts, propose a list of explanations each representing a candidate cluster.

**❷ Assign**: for each sample, assign it to the supporting candidate explanations.

**❸ Select**: select clusters such that each sample is "approximately supported once" via integer linear programming

Corpus
- a. *nothing interesting happened*
- b. *enjoying peace of mind after yoga*
- c. *Heart-broken. can't fall asleep*
- d. *rainy day makes me mildly gloomy*
- e. *!!! My gf agreed to marry me!!*
- f. *...*

Goal
*I want to cluster the texts based on their emotions. Generate a list of explanations for candidate clusters.*

Proposer

Candidate Explanations
- *(i) conveys happiness*
- *(ii) conveys boredom*
- *(iii) conveys sadness*
- *(iv) conveys a strong emotion*
- *(v) ...*

Explanation (i): *conveys happiness*
Sample **b**: *enjoying peace of mind after yoga*
Does the explanation support the sample?

Assigner

*Yes* (1)

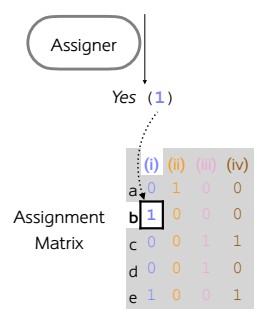

Assignment Matrix

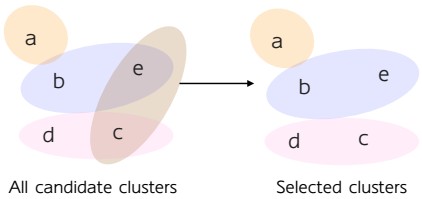

Assignment Matrix (fixed variable)  Selection Vector (optimized variable)  How often each sample is assigned

All candidate clusters → Selected clusters

Figure 2: Our **P**ropose-**A**ssign-**S**elect (PAS) algorithm to tackle the GOALEX task. **Left, Propose**: We prompt a language model ("proposer") with the goal and a subset of the corpus, obtaining a list of explanations for candidate clusters. **Middle, Assign**: we use a language model ("assigner") to determine whether each explanation supports each sample. **Right, Select**: we use integer linear programming to select a subset of explanations, ensuring each sample has roughly one explanation, and obtain the selected set of clusters and explanations as our final output.

for candidate clusters based on the goal and a subset of the corpus. At the assignment stage we address the 2nd desideratum that the explanations should accurately explain the clusters; we do this by assigning text samples only to the explanations that support them. At the selection stage we address the 3rd desideratum on maximizing coverage while minimizing overlap; we do this by using integer linear programming to search for a subset of candidate explanations so that each sample is roughly supported once. At last, we output the selected explanations and their supported samples as clusters.

We benchmarked PAS in two ways: 1) automatically evaluating its ability to recover known clusters from corpora (Section 4), and 2) manually evaluating its clusters and explanations on open-ended corpora (Section 5). For automatic evaluation, we first compared PAS to prior methods on recovering topic clusters underlying news and DBPedia articles and found that PAS is competitive while additionally providing accurate explanations. To test whether PAS is goal-driven, we used an LM to synthesize a corpus, SYNGOALEX, where each text has three known attributes: topic, language, and style; PAS effectively adapts to different goals such as "*clustering by topic / language / style*", while prior methods fail catastrophically.

For open-ended evaluation, we constructed OPENGOALEX, a collection of 12 open-ended

GOALEX problems from various NLP papers. We compared PAS to previous clustering methods such as LDA and found that PAS's explanations are more accurate and goal-related under human evaluation. Finally, we applied PAS hierarchically to create progressively finer-grained clusters on OPENGOALEX, inducing taxonomies over debate arguments, model errors, and customer reviews. [1]

Our contributions are summarized as follows.
- We introduce GOALEX, a novel setting for text clustering that takes into account of user's objectives and provides explanations for each cluster.
- We developed the Propose-Assign-Select (PAS) algorithm and showed its effectiveness on established benchmarks.
- We tested GOALEX to categorize debate points, customer feedback, and model inaccuracies in a hierarchical manner to show its potential to help users navigate extensive datasets effectively.

## 2  Defining GOALEX

We formalize the input-output space of GOALEX and introduce the desiderata for an output.

### 2.1  Input-Output Space

The **input** of GOALEX constitutes
- a set of texts $X$ (the corpus);

---

[1]The implementation and data are made public in https://github.com/ZihanWangKi/GoalEx.

- a string $g$ (the goal description);
- an integer $K$ (the desired number of clusters).

The **output** of GOALEX constitutes

- $K$ strings $e_k, k \in [K]$, where $e_k$ is an explanation of a cluster; additionally $e_k$ needs to be a natural language predicate that can be evaluated against an individual text sample;
- $K$ subsets of $X$: $C_k \subseteq X, k \in [K]$; each representing a cluster.

Note that goals and explanations can be arbitrary natural language strings and predicates much more complicated than the ones in Figure 1. See examples in Section 5.

## 2.2 Desiderata

We list three desiderata for a GOALEX output, which inform our algorithm design in Section 3.

**Goal-Related.** The explanations should be goal-related. For example, if a user's goal is to cluster based on sentiments, then "*has a positive sentiment*" is goal-related, while "*is about sports*" is not.

**Accurate Explanation.** Since each explanation is a predicate, it should have a True evaluation on all samples from its corresponding cluster and False on others. This automatically enforces the clusters to be semantically coherent, where the coherent interpretation is the explanation.

**Minimal Overlap and Maximal Coverage.** The clusters should overlap minimally while their union should cover most of the corpus samples. Ideally, every sample belongs to exactly one cluster.

## 3 The Propose-Assign-Select Algorithm

Each section between 3.1 and 3.3 describes one stage of the PAS algorithm (outlined in Figure 2).

## 3.1 Propose Explanations for Clusters

The proposal stage aims to generate a list of $J$ (around 30-50) candidate explanations $\epsilon_j$.[2] We obtain them by prompting a language model (LM), which we refer to as the "*proposer*" in the remaining text, to perform "in-context clustering" based on a random subset of the corpus; concretely, the prompt concatenates $T$ samples from $X$, the goal $g$, and a request to generate $J'$ explanations for candidate clusters:

*Sample 1. $x_1$; ...... Sample T. $x_T$;*
*Goal: g;*

---

[2]We use $\epsilon_j$ to denote candidate explanations, while using $e_k$ to denote the final selected explanations

*Generate a list of $J'$ explanations for candidate clusters based on the samples.*

where we typically set the maximum $T$ such that the prompt length does not exceed the 75% of the proposer's context window size and $J' = 8 \ll T$. The proposer would respond with a structured list of $J'$ candidate explanations:

*Explanation 1. $\epsilon_1$; ...... Explanation $J'$. $\epsilon_{J'}$.*

Figure 2 left shows a more illustrative prompt-response pair. Since the proposer's context window is usually not long enough to contain the entire corpus, we construct multiple prompts with different subsets from $X$ to allow the proposer to "see" as many different samples as possible during the proposal stage. We sample from the proposer based on different prompts until obtaining $J$ explanations in total. The full prompt is included in Appendix A.

## 3.2 Assign Samples to the Correct Clusters

The assignment stage aims to determine whether each sample $x \in X$ is supported by each explanation $\epsilon_j$. We determine this automatically by prompting an LM, which we refer to as the "*assigner*":

"*Predicate: $\epsilon_j$. Text: $x$.*
*Is the Predicate true on the Text? Yes or No. When uncertain, output No.*"

We therefore obtain an *assignment matrix* $\mathcal{A} \in \{0,1\}^{|X| \times J}$, where $\mathcal{A}_{xj}$ indicates whether $x$ is supported by the $j^{\text{th}}$ explanation.[3] Denote a candidate cluster as $C'_. \subseteq X$, the $j^{\text{th}}$ candidate cluster is thus

$$C'_j := \{x | x \in X, \mathcal{A}_{xj} = 1\} \qquad (1)$$

## 3.3 Select an Optimal Subset of Clusters

The selection stage aims to choose a subset of $K$ clusters from $J$ candidate clusters $C'_j$, so that each sample $x$ belongs to roughly one selected cluster.

Define the selection vector $s \in \{0,1\}^J$ to be a row vector, where $s_j$ indicates whether $C'_j$ is selected. Since we require $K$ selected clusters, we add the constraint:

$$s \cdot \mathbf{1} = K \qquad (2)$$

We introduce a row vector variable $m$

$$m := \mathcal{A}s^T \in \mathbb{N}^{|X|}, \qquad (3)$$

where $m_x$ counts how many selected clusters include $x$. An ideal $s$ should result in $m_x = 1$ for all

---

[3]For convenience we also use $x$ as an index.

$x$, since $m_x > 1$ implies that at least two selected clusters overlap on $x$ while $m_x < 1$ implies $x$ is "missed" by all clusters. Therefore, we design the following loss function $f_\lambda$ to track how much an entry from $m$ diverges from 1:

$$f_\lambda(m_x) := \begin{cases} (1 - m_x) & \text{if } m_x < 1, \text{"miss"}; \\ 0 & \text{if } m_x = 1, \text{"ideal"}; \\ \lambda(m_x - 1) & \text{if } m_x > 1, \text{"overlap"}; \end{cases}$$
(4)

where $\lambda$ is a hyper-parameter determining how much overlaps are penalized. To conclude, we will minimize the following loss $\mathcal{L}$ for $s$

$$\mathcal{L}(s) := f_\lambda(m) \cdot \mathbf{1},$$
(5)

subject to the constraint of Equation 2 and 3.

However, it is hard to directly minimize this loss as written because it requires searching over discrete variables under a piecewise-linear loss. Therefore, we reduce it to an integer linear programming (ILP) problem, which can be effectively solved[4] by existing libraries. To perform the reduction, we introduce an auxiliary row vector variable $a \in \mathbb{R}^{|X|}$ and add the following two constraints

$$a \succcurlyeq 1 - m, a \succcurlyeq \lambda(m - 1),$$
(6)

where $\succcurlyeq$ denotes element-wise greater or equal to. To conclude, we will minimize the loss $\mathcal{L}$

$$\mathcal{L} = a \cdot \mathbf{1},$$
(7)

subject to the constraints in Equation 2, 3, and 6, which are all linear. We explain our implementation in python code with comments in Appendix B. We refer to one sequential application of propose, assign, and select as one *iteration* of PAS.

In addition to the three stages above, PAS involves other auxiliary procedures such as 1) running PAS for 5 iterations to cover the entire corpus, and 2) committing each sample to one single cluster when needed. Due to space constraints, we outline other auxiliary steps of PAS in Appendix D.

## 4 Automatic Evaluation

Following the evaluation protocol from prior works, we evaluated PAS by applying it to corpora that are mixtures of known clusters, treating the known clusters and their explanations as the reference solutions, and checking how well the outputs of PAS

---

[4]efficiently find a reasonable solution empirically even though it is theoretically NP-Complete

can recover the references. We evaluated PAS under two settings: traditional topic clustering and goal-driven non-topic clustering. In both settings, we compared 1) the similarity between the reference and the output clusters automatically and 2) the similarity between the explanations manually. We found that PAS is comparable to previous methods for topic clustering and recovers most of the reference explanations; additionally, since PAS is goal-driven, it performs significantly better when there are multiple ways to cluster a corpus.

The following sections will present the datasets (Section 4.1), the clustering methods we evaluated (Section 4.2), the evaluation protocol (Section 4.3), and the performance of each method (Section 4.4). In addition, we evaluated the quality of each stage of PAS in Appendix F.

### 4.1 Datasets

We evaluated on both corpora from prior works for topic clustering and other corpora for non-topic clustering. We considered four datasets: (AG)'s News, (DB)pedia, (NYT) News, and (SYN)GOALEX. We use (·) to denote a dataset abbreviation in this section.

**(AG)**'s News (Zhang et al., 2015) is a news topic classification dataset with four topic clusters: politics, sports, business, and technology.

**(DB)**pedia (Zhang et al., 2015) is a corpus of articles classified into ontologies, such as athlete and book, with 14 topic clusters in total.

**(NYT)** News (Meng et al., 2020) is a corpus of New York Times news articles, each with a topic label and a location label. There are in total 9 topics, e.g., politics, arts, and 10 locations, e.g., France, Italy. We subsampled this corpus so that the topic and location labels are balanced.

**(SYN)**GOALEX To test PAS's ability to cluster based on different goals, we synthesized (SYN)GOALEX, which can be clustered based on three different dimensions: Topics, Writing Style, or Language. To synthesize SYNGOALEX, we first designed four values for each dimension, e.g. "1.*French*"/2."*English*"/3."*Spanish*"/4."*Deutsch*" for the Language dimension. Then we took the Cartesian product across three dimensions, obtaining $4^3$=64 value combinations; for example, one combination could be "*Language: French, Style: Poem, Topic: Sports*". Finally, for each of the 64 value combinations, we prompted Claude-v1.3 to generate 16 text samples condi-

tionally on the values, resulting in 1024 samples for SYNGOALEX in total. Therefore, the reference clusters are different if we cluster based on different dimensions, hence penalizing methods that ignore the goals. Appendix C includes more details about the values for each dimension and the prompt we used for conditional generation.

The first three datasets might have appeared in the pre-training corpus of `gpt-3.5-turbo`, thus raising potential concerns about memorization. We believe our task of proposing explanations on the three datasets did not occur in the pre-training corpus, thus justifying the validity of our evaluations. A more detailed justification is in Appendix E.

### 4.2 Methods and Baselines

We compared fours methods: PAS, LDA, E5, and Instructor. For all methods, we set the number of clusters to be that of the reference solution. We use $\underline{\cdot}$ to denote a method in this section.

**PAS** is described in Section 3. By default, we used `gpt-3.5-turbo` as the proposer and `flan-t5-xl` (Chung et al., 2022) as the assigner. We set $J = 30$ and $\lambda = 0.5$, except for the (DB)pedia dataset and (NYT) News dataset where we set $\lambda = 0.3$ since they have many target clusters. We additionally require each $x$ to appear in exactly one cluster using the commitment method described in Section D.

**LDA** (Blei et al., 2003), or Latent Dirichlet Allocation, is a standard generative probabilistic model that identifies hidden topic clusters in a corpus by assuming that each text is a mixture of topics and each topic is a distribution of words.

**Instructor** (Su et al., 2022) is contrastively trained on a large collection of datasets with annotated instructions; as a result, it can create specialized text embeddings according to the instructions. To perform goal-driven clustering, we rephrased our goal as the embedding instruction. We computed the text embeddings with `instructor-xl` and then ran K-means clustering.

**E5** (Wang et al., 2022) is a contrastively trained text embedder on crawled data, e.g. post-comment pairs and annotated data, e.g. NLI. We computed the text embeddings with `e5-large` and then ran K-means to obtain the clusters.

Appendix L includes more implementation details, e.g. what library we used.

### 4.3 Metrics

We follow the standard protocol from Lange et al. (2004) to evaluate a clustering method, where we

| Macro F$_1$ (%) | (AG) | (DB) | (NYT) | (SYN) |
|---|---|---|---|---|
| Random | 27 | 11 | 14 | 27 |
| LDA | 53 | 51 | 51 | 28 |
| E5 | 86 | 72 | 67 | 96 |
| Instructor | 84 | **82** | 69 | 77 |
| PAS | **87** | 71 | **70** | **98** |

Table 1: We compare different methods and PAS for recovering topic clusters and report the macro F$_1$ score for each method, along with a random baseline which assigns each sample to a cluster randomly.

| label | output | F$_1$ |
|---|---|---|
| " *Company*" | " *business*" | 75 |
| " *Building*" | " *architecture*" | 82 |
| " *Animal*" | " *lakes*" | 3 |
| " *Plant*" | " *biology*" | 73 |
| . . . | . . . | . . . |

Table 2: We ran PAS on (DB)pedia to cluster based on topics and present its cluster explanations. We abbreviate each explanation by removing the prefix "*has a topic of*" (e.g., " *Company*" corresponds to a full explanation " *has a topic of Company.*"). The table of all 14 explanations is in Appendix Table 15.

first match each of the output cluster to a known reference cluster and then compute the similarity of each pair of matched clusters via F$_1$ score. Denoting the $k^{\text{th}}$ output cluster as $\hat{C}_{k'}$ and the $k^{\text{th}}$ reference as $C_k^*$, we formulate the matching problem as a bipartite matching problem and solve it with Hungarian algorithm, where the edge weight between each pair of reference and output cluster is the size of their overlap, $|\hat{C}_{k'} \cap C_k^*|$. After matching finishes, for each pair of matched reference and output clusters $C^*$ and $\hat{C}$, we compute the F$_1$ score of predicting whether $x \in C^*$ based on whether $x \in \hat{C}$, and then average across all $k$ to compute the final macro F$_1$ score for evaluation.

### 4.4 Results

We evaluated PAS on topic clustering and goal-driven clustering based on other dimensions. All results shown below are the average of 3 trials with different random seeds.

**Recovering Topic Clusters.** We first evaluated the clustering methods on recovering topic clusters and report the results in Table 1. PAS consistently outperforms LDA. PAS slightly outperforms Instructor on (AG), (NYT), and (SYN); on (DB), PAS is underperforming Instructor by 11%.

To understand why our method does not deliver the best performance on (DB), we manually exam-

| Macro $F_1$ (%) | (NYT) | (SYN) | |
|---|---|---|---|
| | Location | Language | Style |
| Random | 13 | 27 | 27 |
| LDA | 40 | 83 | 25 |
| E5 | 55 | 27 | 27 |
| Instructor | 54 | 25 | 25 |
| PAS | **76** | **97** | 31 |
| PAS† | - | - | **45** |

Table 3: We ran PAS to cluster based on non-topic goals and report the macro $F_1$ score. † We use `gpt-4` as the proposer and `gpt-3.5-turbo` as the assigner.

ined the explanation (for one of the three trials) for each output cluster and present it in Table 2 along with its matching reference. Overall, the outputs are similar to the references; the performance drop is mainly because PAS completely missed the "*Animal*" cluster, since it "merged" it with the "*Plant*" reference cluster into a "*Biology*" cluster. Additional evidence on this merging effect can be found in Appendix F. We consider such a mistake benign and hence conclude that PAS is on par with previous state of the art on topic clustering.

Next, we will show that Instructor fails catastrophically on non-topic based clustering, implying that it has an "inductive bias" to cluster on topics.

**Recovering Non-Topic Clusters.** We now evaluate PAS on other goals and report the performance in Table 3 — specifically, clustering based on locations on (NYT) and writing styles or languages on (SYN). Since PAS is goal-driven, it performs significantly better than previous methods.

However, PAS with our default configuration is poor at writing style clustering on (SYN). Fortunately, the performance can improve significantly by 14% after using more capable models – `gpt-4` (OpenAI, 2023) as the proposer and `gpt-3.5-turbo` as the assigner. We present the output explanations in Table 4 and expect PAS to improve with future better LMs.

**Sensitivity Study.** We conducted a prompt sensitivity study in Appendix M and concluded that our method is not sensitive to the prompts we chose for the proposer and assigner. We also conducted a dataset sensitivity study on (DB)pedia to study how imbalance of classes in the dataset or noisy, out-of-distribution data points would affect our algorithm, and concluded that our algorithm is not especially vulnerable to these noises than Instructor.

**Ablation Studies for PAS.** Finally, we conducted two ablations for PAS to study the contribution of 1) proposing multiple iterations and 2) our selection algorithm. We present the results in Appendix

G and found that running PAS for five iterations improves over one iteration and using an ILP algorithm with a positive $\lambda$ improves the performance.

## 5 Open-Ended Advanced Applications

While PAS achieves high performance on benchmarks with cluster labels, it does not necessary imply high performance under real applications. Therefore, we constructed OPENGOALEX, a collections of 12 open-ended realistic GOALEX problems to evaluate PAS. Since these problems do not have cluster labels, we evaluated PAS with the three metrics introduced in Section 2: (1) explanation accuracy, (2) goal-relevance, and (3) coverage and overlap. As (1) and (2) require human annotators and are hence expensive to conduct repeatedly, we used them to test the limit of PAS to inform future research: we applied PAS with the highest quality models under our budgetary constraints and challenged it to generate taxonomies by producing trees of progressively finer-grained clusters on OPENGOALEX. We evaluated PAS quantitatively with human annotators for the first layer of the taxonomy and qualitatively analyzed the rest.

### 5.1 OPENGOALEX

To evaluate under real applications, we constructed OPENGOALEX, a collection of 12 open-ended GOALEX problems. Each corpus comes from an NLP paper or a Kaggle website and we annotated it with a goal related to the paper. For example:

- comments for model-generated summaries (Scheurer et al., 2023), with the goal of "*categorizing model errors*
- debates on why spanking is bad, with the goal of "*categorizing the types of arguments*" (Habernal and Gurevych, 2016)

Appendix I includes all 12 problem descriptions and citations. To reduce reporting bias, we collected OPENGOALEX before our experiments.

### 5.2 Advanced Application of PAS

To generate a taxonomy for each corpus, we first apply PAS for the entire corpus; then for every output cluster with $> 20$ samples, we apply PAS again to create finer-grained clusters and output trees of explanations as taxonomies; when creating child clusters for a parent cluster, we include the explanation for the parent into the original goal and request the new candidates to be sub-categories of the parent's explanation. Here is an example goal

| reference | output | F$_1$ |
|---|---|---|
| *"has a writing style of twitter"* | *"has a writing style of instructional or informat..."* | 45 |
| *"has a writing style of screen play"* | *"has a writing style of narrative or storytelling..."* | 51 |
| *"has a writing style of rap"* | *"has a writing style of using rhymes and rhythm"* | 49 |
| *"has a writing style of poem"* | *"has a writing style of incorporating foreign lan..."* | 34 |

Table 4: We ran PAS with proposer=gpt-4 and assigner=gpt-3.5-turbo to cluster (SYN)GOALEX based on Style. We present the four output explanations and compare them to the references. Although our method is still far from perfect, the PAS is able to generate similar explanations for the 2nd and 3rd row.

Figure 3: Explainability evaluation instances. Given the sample $x$, the evaluator needs to choose which one of the explanations (blue, model generated explanation for $x$, or orange, a random different model generated explanation) is more related. **Left**: top-word based explanations by LDA and Instructor. **Right**: natural language explanations by PAS.

where the parent's explanation is in **bold**:
*"My goal is to cluster comments for model-generated summaries falling under the following category: **whether this comment advises adding omitted details; specifically, . . . . For example, . . .** ', and I want to create finer-grained cluster descriptions that fall under the above category."*

We set $K = 8$, $\lambda = 0.5$, proposer $=$ gpt-4, and assigner $=$ Claude-v1.3 (Bai et al., 2022). We allow a sample to appear in multiple clusters so that PAS can see as many samples as possible when creating subcategories. We designed a new prompt template to propose more detailed explanations; see Appendix Figure 9 for more details.

### 5.3 Quantitative Evaluation

We quantitatively evaluated the first layer of taxonomy (i.e. the output of the standard GOALEX formulation) based on the three metrics introduced in Section 2. To help the readers interpret our results, we compared PAS to LDA and Instructor.

**Explanation Accuracy.** If an explanation $e_k$ is accurate for cluster $C_k$, then given a sample $x \in C_k$ in its cluster, a human should be able to tell whether $e_k$ or $e_{k'}$, the explanation for another cluster $C_{k'}$, is more related. We call the tuple $(x, e_k, e'_k)$ an explainability evaluation instance and show an example in Figure 3. To sample an instance, we randomly sampled a problem from OPENGOALEX, sample an output cluster $C_k$, and then sampled a text sample $x \in C_k$; we then randomly sample a

distractor explanation $e_{k'}$ such that $x \notin C_{k'}$. For each instance, we present it to three human turkers and consider it correct if the majority of them choose $e_k$ over $e_{k'}$. We include more details for this HIT task and how to generate word-based explanations for LDA and Instructor in Appendix J.

We ran study with Turkers and found that they can choose the corresponding explanation 80% of the time for PAS, outperforming 56% for LDA ($p \approx 10^{-9}$) and 71% for Instructor ($p < 10^{-3}$).[5]

**Relevance.** We evaluated how well PAS's explanations relate to the goal and compared them to explanations for LDA and Instructor clusters. For each problem in OPENGOALEX, we randomly sampled a problem, an explanation from PAS's output, and one from a baseline approach; we then asked the evaluators to choose which explanation is more relevant to the goal, or abstain if they are similar. To ensure reliability and fairness of our evaluation, the authors performed evaluations on their own rather than relying on Turkers, since the goals in OPENGOALEX are technical and motivated by NLP research; the evaluators are also unaware of whether the baselines or PAS generated each explanation: to make the baseline explanations stylistically similar to PAS's outputs, we used the D5 system by Zhong et al. (2023) to describe the differences between each cluster and the rest of the corpus in natural language.

Table 5 reports the results for a direct pair-wise comparisons between PAS and LDA/Instructor. PAS's explanations are more often related to the goal compared to LDA ($p$-value $< 10^{-3}$) and Instructor ($p$-value $< 0.05$), which are not goal-driven. As a robustness check, two authors independently reproduced the exact same conclusion.

**Coverage and Overlap.** On average, 66% of the text samples are covered by at least one cluster, and

---

[5]Our evaluation weighted each cluster explanation uniformly, so these results imply that "PAS produces on average more accurate explanations", but not "each sample $x$ is more accurately explained".

| Baseline | Win (%) | Lose (%) | $p$-value |
|---|---|---|---|
| Instructor | 30 | 12 | $< 10^{-3}$ |
| LDA | 51 | 13 | $< 10^{-8}$ |

Table 5: How often is PAS's explanations are more relevant compared to the baselines.

60% of the samples are covered by one and only one cluster; this is one of the key limitations of the current PAS system, as traditional clustering methods such as LDA or Instructor would cover 100% samples exactly once. On the flip side, however, these low numbers might reflect the inherent difficulty of producing semantically coherent clusters with $K = 8$ in a goal-driven way; when a sample cannot be supported by any explanation, it might actually be better to explicitly consider it "not covered" as our approach does, rather than forcing it to a semantically incoherent cluster and creating a delusion of 100% coverage.

### 5.4 Qualitative Analysis

We show one subpart of an example taxonomy of '*"why spanking is bad"* in Figure 4 to obtain a qualitative understanding of PAS. Most explanations are goal-related and they could help the users quickly explore the corpus without inspecting each cluster manually; however, some do not form a coherent taxonomy. For example, the explanation "*employs rhetorical questions*" is irrelevant to the goal of identifying argument types; additionally, the explanation "*discusses the cycle of violence*" appears both in the first and second levels of the taxonomy and hence should be merged. We present example taxonomies over customer complaints and model errors in Appendix Figure 12 and 13.

## 6 Related Work

**Text Clustering.** Most existing text clustering methods first encode each text sample into some vector and then run a clustering algorithm; e.g. one hot bag-of-words encodings and tf-idf (Blei et al., 2003; Aggarwal and Zhai, 2012), or neural word/context embeddings (Aharoni and Goldberg (2020b); Wang et al. (2022); Su et al. (2022), *inter alia*). Using text clustering methods as backbones, many prior works such as Luu et al. (2014); Shang et al. (2020); Downey et al. (2015) apply them hierarchically to a corpus to produce taxonomies over topics. Nevertheless, previous text clustering algorithms do not necessarily produce interpretable clusters (Chang et al., 2009), mostly studies topic

clustering, and cannot flexibly adapt to users' goal.

**Explaining Text Clusters.** To explain topic clusters, Carmel et al. (2009) proposes to explain each cluster by extracting candidate labels either from text or from Wikipedia; Treeratpituk and Callan (2006) proposes to explain each cluster by selecting candidate labels using information from the cluster, the parent cluster, and corpus statistics; Zhang et al. (2018) proposes to summarize a cluster with a group of semantically coherent phrases. However, these solutions are limited, since phrase-level explanations are not flexible enough to describe a complex cluster. Zhong et al. (2022) proposes to explain a text cluster by describing its differences with the rest of the corpus in natural language; however, its explanation usually does not fully cover the entire cluster, while our clusters are explainable by construction during the assignment stage.

**Controlling the Clustering Process.** We need additional supervision signals from the users so that they can have more control over the clustering process. Hu et al. (2014) allows the users to shape the clusters by specifying words that should co-occur in a cluster. In the image domain, Open World Classification (OWC) (Shu et al., 2018; Cao et al., 2021), also known as Generalized Category Discovery (Vaze et al., 2022), gives the users more control by asking for a few example labels and their example datapoints; for example, given five labels and some corresponding images in a CIFAR10 dataset (Krizhevsky et al., 2009) (e.g., "automobile", "bird", etc), discover the remaining five labels on the unlabeld dataset (e.g., "frog", "ship") and classify the entire dataset into 10 labels (Zheng et al., 2022; Zhao and Mac Aodha, 2023; Xie et al., 2023); closest to our work, Wang et al. (2023) operates OWC in the text domain. Our work proposes a complementary direction and allows the user to control the clustering process with a goal description, which is more expressive and lightweight.

**Explaining Patterns via Language.** Natural language can be used to help users explain patterns in text data (Zhong et al., 2022; Singh et al., 2022). With the increasing capability of language models (OpenAI, 2023), they are used to explain more complicated patterns, such as the inner workings of neural networks (Singh et al., 2023; Bills et al., 2023). Our system is closest to D5 developed by Zhong et al. (2023), which describes difference between text distributions in a goal-driven way.

Patterns in other modalities can also be described

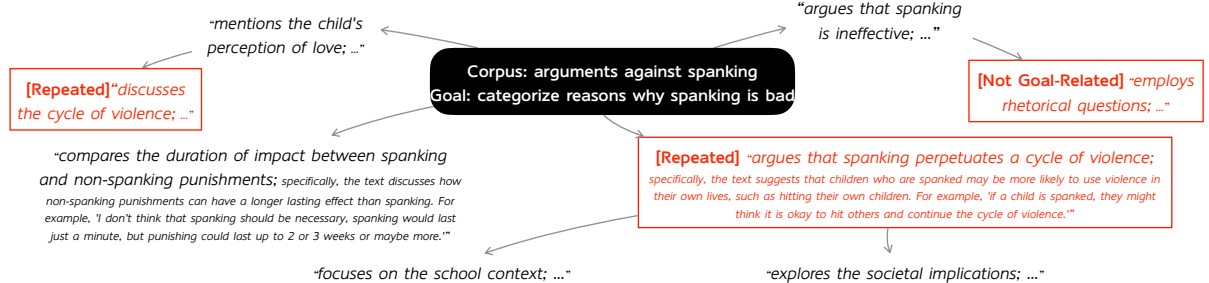

Figure 4: Example taxonomy for arguments against spanking (Habernal and Gurevych, 2016) produced by PAS.

via language. For example, Zhu et al. (2022) describes distribution shifts for image classification tasks and Eyuboglu et al. (2022) describes errors made by vision models. With future advances in multi-modal foundation models, we hope that GOALEX can be extended to cluster images, and potentially sound (Aghajanyan et al., 2023) or physical senses (Thomason et al., 2016).

## 7 Conclusion

We proposed GOALEX, a new formulation for text clustering that adapts to a user's goal and outputs explanations for each cluster. To tackle GOALEX, we developed the Propose-Assign-Select (PAS) algorithm; under automatic evaluation with known references and open-ended applications, PAS can generate accurate and goal-related explanations for the clusters. Finally, we applied GOALEX hierarchically to produce taxonomies over debate arguments, customer complaints, and model errors, thus assisting users to explore large corpora. Future works can improve on discovering minority clusters, following the goal better, and resolving global inconsistency when applying PAS recursively.

## Limitation

As indicated in Section 5.4, PAS cannot yet construct coherent taxonomies. As indicated in Section 5.3, PAS is far from being able to cover all the samples and the clusters have significant overlap. Given these weaknesses, a practitioner should still properly interpret the results of PAS.

Our evaluation is also not universal in scope. Our benchmarks are predominantly in English, and hence our results do no necessarily generalize to other languages. Our dataset OPENGOALEX also implicitly encodes the author's biases for what clustering problems are more important than the other, though this is a universal problem for any newly proposed benchmark. We hope that with a combination of automatic evaluation on datasets from

prior work and human evaluation on open-ended GOALEX problems that we collected, we can more robustly, though not perfectly, establish the conclusions from our paper. We also did not evaluate our methods under situations where the number of clusters $K$ is large, e.g., $K > 50$.

Finally, reaching the best performance requires using gpt-4 and claude-v1.3 as the proposer and the assigner, which might induce a large cost via LM-APIs if one needs to run PAS on a large corpus; we hope such a problem would alleviate in the future if we could use a lighter weight model to approximate the assigner, the cost of computation significantly decreases, or there is a more computationally efficient variant of PAS.

## Ethics Statement

The human evaluation is approved by the Institutional Review Board.

## Acknowledgement

Our work is sponsored in part by NSF CAREER Award 2239440, NSF Proto-OKN Award 2333790, NIH Bridge2AI Center Program under award 1U54HG012510-01, Cisco-UCSD Sponsored Research Project, as well as generous gifts from Google, Adobe, and Teradata. Any opinions, findings, and conclusions or recommendations expressed herein are those of the authors and should not be interpreted as necessarily representing the views, either expressed or implied, of the U.S. Government. The U.S. Government is authorized to reproduce and distribute reprints for government purposes not withstanding any copyright annotation hereon. Ruiqi Zhong is funded by NSF-Simons Theorinet Grant (NSF Award #2031985). We thank members from Jacob Steinhardt's group, Jingbo Shang's group, and Berkeley NLP group for paper feedback.

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

## A   Prompt Templates

**Proposal Stage.** Figure 5 shows the prompt we used on SYNGOALEX to propose simple explanations for the clusters. Figure 7 shows the perturbed prompt for conducting prompt sensitivity analysis. Figure 9 shows the formatting instruction we used on OPENGOALEX to propose more sophisticated explanations for the clusters.

**Assignment Stage.** Figure 6 shows the prompt we used to check whether an explanation supports a sample. Figure 8 shows the perturbed prompt for conducting prompt sensitivity analysis.

**Prompt to Commit to a Single Explanation.**

> *Predicate 0: $\epsilon_1$*
> *Predicate 1: $\epsilon_2$*
> *...*
> *Predicate K: $\epsilon_K$*
> *Text: $x$.*
> *Choose the Predicate the matches the Text the most.*

## B   Selection Stage Implementation

To help the reader understand our algorithm for selecting the descriptions, we include our python implementation in Figure 10 and 11.

## C   Synthesizing SYNGOALEX

We first describe the four values for each dimension and then discuss the prompts we used to generate the text samples in SYNGOALEX.

**Topic**: 1."*has a topic of what sports to do to improve your health*", 2. "*has a topic of a new anime has been announced*", 3. "*has a topic of a tech company releases a new groundbreaking paper*", and 4. "*has a topic of how to improve your productivity*".

**Writing Style** 1. "*has a writing style of twitter*", 2. "*has a writing style of screen play*", 3. "*has a writing style of rap*", and 4) "*has a writing style of poem*".

**Language** 1. "*has a natural language of English*", 2. "*has a natural language of French*", 3. "*has a natural language of Deutsch*", and 4) "*has a natural language of Spanish*".

To generate the text samples conditioned on three dimensions, we first generated "content samples" based on the topic. For each topic, we first asked GPT-4 OpenAI (2023) to generate 40 English news summary for each topic using as diverse vocabulary as possible. Here is the prompt template we used, where we substituted {topic} with the topic we want to condition on.

> "*Write 40 news 2-sentence paragraphs about topic {topic}, using as diverse vocabulary as possible. We prefer being concrete; e.g. we prefer 'James Lebron joins Lake to strengthen the team; how likely is the team going to win next time?', rather than genric statements "it's common to recruit new team members". Additionally, you cannot use words that are directly used in the topic.*
>
> *The list continues below.*"

For each of the 64 value combinations, we first sampled 16 text samples from the content samples based on its topic, and then for then prompted Claude-1.3 (Bai et al., 2022) to rewrite it with a different style and language. The template we used is as follows, where we substituted {original_text} with the text to be rewritten, and {style} and {language} to be conditioned on:

> "*{original_text}*
>
> *Rewrite the above paragraph in the style of {style} in {language}.*"

## D   Other Techniques Used by PAS

**Multiple Iterations of PAS.** As mentioned in Section 3.1, since the proposer's context window might not be long enough to contain the entire corpus, the

```
In this task you will need to come up with categories to group the text
together based on a goal, where each category can be described with a natural
language predictate. For example

1. How are you doing?
2. How do I apply for visa?
3. Isn't it ridiculous?
4. How to get there by bus?
5. How is it going?
6. Aren't technological advancement good for the humanity?

Here is our goal: I want to cluster the questions based on their
functionalities. Your responses are:
– "is a rhetorical quesiton"
– "is meant for greetings"
– "is asking for practically useful information"

(note that all of them need to be natural language predicates that can be
validated on individual samples)

Here are some texts:
                        x_1, x_2, x_3 .... x_T
{samples_in_prompt}
                      g
Here is our goal: {goal} Please suggest me at most
{num_descriptions_per_prompt} descriptions, one in a line, starting with "–"
                      K
and surrounded by quotes"". In terms of formatting, each of them needs to be a
predicate about a text, for example:
– "uses double negation"
– "has a conservative stance"

Do not output anything else. (Note that the examples might not be goal related,
and your response should be both formatted correct as above and related to the
goal.)
{example_description_in_prompt}

Again, here's the goal: {goal}. Your responses are:
– "
```

Figure 5: The template we used to propose candidate explanations, where we will substitute the corresponding variables to construct the prompt.

```
Check whether the TEXT satisfies a PROPERTY. Respond with Yes or No. When
uncertain, respond with No. Do not say anything else.

Now complete the following example –
input: PROPERTY: {predicate}  ε
TEXT: {text}    x
output:
```

Figure 6: The template we used for the assigner which decides whether a candidate explanation supports a text sample.

In the given task, you are required to devise categories to classify the texts according to a specific objective. Each category should be expressible via a natural language predicate. Here's an example:

0. How are you doing?
1. How do I apply for a visa?
2. Isn't it ridiculous?
3. How to get there by bus?
4. How is it going?
5. Aren't technological advancements good for humanity?

The objective here is: I am looking to categorize these questions based on their purpose. Your response could be:
- "is a rhetorical question"
- "is intended for pleasantries"
- "is asking for practical information"

(remember these responses must be written as natural language predicates which can be verified against the provided sample texts)

Here are some text samples:

{samples_in_prompt}

The objective is: {goal}
Please can you provide a number of descriptors, beginning with "-" and are within "". For formatting purposes, every response must be a predicate about a text, for instance:
- "employs double negation"
- "exhibits a traditional standpoint"

Do not provide any other outputs. (Note that these examples might not correspond to the actual objective, yet your response should both adhere to the above-mentioned formatting and be pertinent to the stated goal.)
{example_description_in_prompt}

Once more, the objective is: {goal}. Your responses could be:
- "

Figure 7: The perturbed proposal template.

Determine if the TEXT adheres to a certain PROPERTY. Give a Yes or No response. When in doubt, lean towards No.

Now, proceed with the following example -
input: PROPERTY: {candidate_explanation}
TEXT: {text}
output:

Figure 8: The perturbed assigner template.

```
Here is our goal: {goal} Please suggest me at most
{num_descriptions_per_prompt} descriptions, one in a line, starting with "-"
and surrounded by quotes"". In terms of formatting, each of them needs to be a
predicate about a text, followed by an explanation and an example, for example:
- "uses double negation; specifically, there is a sentence in the text that
uses negation twice. For example, 'the pilot did not never had an accident'"
- "has a conservative stance; specifically, the overall text exhibits a
conservative political stance (e.g. pro-life, deregulation, etc). For example,
'Fetus is sentient so we should not abort.'"
```

Figure 9: We used the same template to propose hypotheses in Section 5, except that we changed the formatting instruction to propose more sophisticated predicates. The changed part is shown above and the key changes are underlined in red.

```
import numpy as np
from typing import List
import pulp

def select_description_indexes(
    assignment_matrix: np.ndarray, overlap_penalty: float, num_clusters: int
) -> List[int]:
    """
    Select the indexes of the descriptions s.t. they cover the entire dataset
    approximately once.

    Parameters
    ----------
    assignment_matrix : np.ndarray
        The assignment matrix of the descriptions. The shape is
        (num_datapoints, num_descriptions). 1 if the description supports the
        datapoint, 0 otherwise.
    overlap_penalty : float
        The penalty for the overlap of the descriptions. \lambda in the paper.
    num_clusters : int
        The number of descriptions to select. K in the paper.

    Returns
    -------
    List[int]
        The indexes of the selected descriptions.
    """
```

Figure 10: The function signature of the selection stage. The function body can be seen in Figure 10

proposer might never "see" some samples or their similar variants; consequently, some samples might not be supported by any of the proposed explanations. Therefore, after one iteration of PAS, we collect all samples not supported by any explanation and use them to propose candidate explanations again, hoping that some of the new candidates will support them. To ensure broad supports over the entire corpus, we ran PAS for 5 iterations.

**Commit to a Single Cluster.** At the end of PAS, some samples might be supported by multiple selected explanations. However, the user or a benchmark might require each sample to commit to one single most appropriate cluster. Consider the following sample in a news corpus: $x$="...*after years of disputes over security and national pride, Germany and the United_States signed agreed to build a new American embassy on the empty lot* ...". $x$ is supported by both explanations $\epsilon_1$="*is related to real estate development*" and $\epsilon_2$="*is related to international politics*". While both $\epsilon_1$ and $\epsilon_2$ literally support $x$, $\epsilon_2$ is more appropriate. To commit each sample to one single cluster, we prompted an LM with all selected explanations and asked it to choose one of them as our final commitment. See

our prompt template in Appendix A.

# E  Potential Effects of Memorization

The leakage of test data might affect both the proposer `gpt-3.5-turbo` and the assigner `flan-t5-xl` based on T5 (Raffel et al., 2020).

For the proposer, since the prompt represents a novel usage of large language model, we do not expect the proposer prompt specifically based on (AG), (DB), and (NYT) to have appeared in the pre-training corpus. That said, it is plausible that there are similar texts which produce categories based on other corpus, and they might have improved LM's capability to perform "in-context clustering"; on the other hand, however, such a consideration is irrelevant since we are not claiming that `gpt-3.5-turbo` can perform "in-context clustering" zero-shot without any similar training data. Finally, PAS recovers two of the uncommon topic labels ("*anime*" and "*productivity*") for (SYN), produces reasonable explanations not identical to the reference on (DB), and generates novel explanations for the clusters in Section 5; these empirical evidence suggests that the capability of the proposer is largely not due to memorizing the training

```python
# Create the pulp optimization problem
problem = pulp.LpProblem("TextCoveringProblem", pulp.LpMinimize)
num_datapoints, J = assignment_matrix.shape

# the selection vector we will optimize, 1 if the description is selected,
# 0 otherwise
s = [pulp.LpVariable(f"s_{j}", cat="Binary") for j in range(J)]

# how often each datapoint is covered by the selected descriptions
m = [
    pulp.LpVariable(f"m_{x}", 0, None, cat="Continuous")
    for x in range(num_datapoints)
]

# the penalty for each datapoint, an auxiliary variable
a = [
    pulp.LpVariable(f"a_{x}", 0, None, cat="Continuous")
    for x in range(num_datapoints)
]

# Define objective function
problem += pulp.lpSum(a)

# Constraint: the number of selected descriptions is num_clusters
problem += pulp.lpSum(s) == num_clusters

for x in range(num_datapoints):
    # adding the constraint that m[x] = the number of descriptions that
    # cover datapoint x
    problem += m[x] == pulp.lpSum(assignment_matrix[x, j] * s[j] for j in
    range(J))

    # adding the constraint that z[x] >= overlap_penality * (m[x] - 1)
    problem += a[x] >= overlap_penality * (m[x] - 1)
    # adding the constraint that z[x] >= not_cover_penality * (1 - m[x])
    problem += a[x] >= (1 - m[x])

# Solve the problem
pulp.LpSolverDefault.msg = 0
problem.solve()

# Extract the solution based on the selection vector
selected_idxes = [j for j in range(J) if pulp value(s[j]) == 1]
```

Figure 11: The function body of the selection stage. The function signature can be seen in Figure 10

data.

As for the assigner, it has a similar functionality as classification or entailment. Since both of these tasks are already relatively straightforward for state-of-the-art language models (Gilardi et al., 2023), we do not consider the potential effect of memorization on (NYT), (DB), and (AG) to play a significant role in our evaluation.

# F  Per-stage Evaluations

In addition to an automatic end-to-end evaluation discussed in Section 4, we also present an automatic *per-stage* evaluation to better understand the quality of each stage of the PAS algorithm.

**Assign Stage.** We first evaluate the second Assign stage, where we try to understand how well the assigner can recover the clusters when given the reference explanations of each cluster. This evaluation will measure the unavoidable discrepancy that might be brought by an imperfect assigner, such that later evaluations for other stages can marginalize out this discrepancy. We formalize the evaluation as following: Recall that we denote the corpus as $X$. Let $\overline{C_k}$ be the set of texts that belong to class k and $C_k$ the set of texts that are supported by the reference explanations of the class. We define the recall as

$$\frac{|\overline{C_k} \cap C_k|}{|\overline{C_k}|} \quad (8)$$

and the specificity as

$$\frac{|X - \overline{C_k} \cap X - C_k|}{|X - \overline{C_k}|} \quad (9)$$

Then, we take the average of recall and specificity as the score for this class and the average score over all classes as the score for the assigner on the dataset. The reason for analyzing recall and specificity is that recall is invariant if we found a superset of a class (e.g., a parent class in a hierarchy) and specificity is invariant when we find a subset of a class (e.g., a child class in a hierarchy). For the assigner, we expect the exact clusters to be found, hence we consider both recall and specificity. In latter two evaluations, since our method can operate in a top-down hierarchical manner, we focus on the recall, but not the specificity.

In Table 7, we illustrate the results for the assigner (and the later evaluations) on the two harder datasets ((NYT) Topics and (DB)pedia) where our method achieves a reasonable, but not perfect per-

| reference explanation | recall | specificity |
|---|---|---|
| " *Company*" | 44 | 100 |
| " *Educational Institution*" | 63 | 100 |
| " *Artist*" | 26 | 100 |
| " *Athlete*" | 99 | 100 |
| " *Office Holder*" | 92 | 100 |
| " *Mean Of Transportation*" | 64 | 100 |
| " *Building*" | 35 | 100 |
| " *Natural Place*" | 41 | 97 |
| " *Village*" | 87 | 100 |
| " *Animal*" | 76 | 100 |
| " *Plant*" | 91 | 100 |
| " *Album*" | 46 | 100 |
| " *Film*" | 64 | 100 |
| " *Written Work*" | 28 | 100 |
| **Assigner Score** | | 80 |

Table 6: Assign-stage evaluation on (DB)pedia of `flan-t5`. We abbreviate each explanation by removing the prefix "*has a topic of*" (e.g., " *Company*" corresponds to a full explanation " *has a topic of Company.*").

| Dataset | Assign Score |
|---|---|
| (DB) | 80 |
| (NYT) | 73 |

Table 7: Assign-stage evaluation on (DB)pedia and (NYT) Topics of `flan-t5`.

formance. Table 6 contains a per-explanation example for (DB)pedia.

**Propose Stage.** In the Propose Stage, we would like to understand how well the proposed explanations capture the reference explanations. Therefore, for each reference explanation, we look into the proposed description that has highest recall to the reference explanation, as our method can operate in a top-down hierarchical manner to identify more specific clusters. We average the highest recall for each reference explanation as the score for the proposer.

To remedy variances and get better understanding of the propose stage, we ask the proposer to make a large number of proposes (we used 128 proposes for both datasets) and repeat the experiment three times with different random seeds; the random seeds vary the text that is given to the proposer, therefore, could yield different proposed explanations from the proposer. The results are presented in Table 9. We additionally show the number of proposed explanations that are actually matched (i.e., that are the highest recall for some reference explanations). Table 8 contains a per-explanation example for (DB)pedia. Overall, the proposer has

| reference explanation | proposed explanation | recall |
|---|---|---|
| " *Company*" | " *history*" | 46 |
| " *Educational Institution*" | " *history*" | 56 |
| " *Artist*" | " *entertainment*" | 79 |
| " *Athlete*" | " *sports*" | 99 |
| " *Office Holder*" | " *politics and ...*" | 87 |
| " *Mean Of Transportation*" | " *history*" | 74 |
| " *Building*" | " *history*" | 69 |
| " *Natural Place*" | " *geography*" | 97 |
| " *Village*" | " *geography*" | 98 |
| " *Animal*" | " *zoology*" | 80 |
| " *Plant*" | " *botany*" | 96 |
| " *Album*" | " *entertainment*" | 60 |
| " *Film*" | " *entertainment*" | 100 |
| " *Written Work*" | " *entertainment*" | 76 |
| **Proposer Score** | | 80 |

Table 8: Propose-stage evaluation on (DB)pedia of `gpt-3.5-turbo`, the used reference assigner is `flan-t5`.

| Dataset | Propose Score | # matched proposes |
|---|---|---|
| (DB) | 80 | 7 |
| (NYT) | 86 | 9 |

Table 9: Propose-stage evaluation on the 14-class (DB)pedia and 9-class (NYT) Topics of `gpt-3.5-turbo`. The score and the # of matched proposes are averaged over three runs.

a high coverage over the reference explanations. For (DB)pedia, the number of matched proposes is one half of the true number of classes, likely due to general explanations matched (e.g., " *Album*" and " *Film*" both are matched by " *entertainment*").

**Select Stage.** In the Select Stage, we are also interested in how well the proposed explanations cover the desired reference explanations. We again use the average highest recall as the score. In Table 11 we show the score for the select method in PAS. We note that there is a drop in coverage (i.e., a drop from propose score to select score) during the select phase, even though it is potentially possible to pick exactly all the matched proposes in the proposer. This indicates a potential room of improvement for the select algorithm.

**Clustering w/o Cluster Number Constraints.** We would like to point out that our select algorithm does not have to enforce a number of clusters[6]. We could have remove the constraint of number of clusters and add an penalty proportional to the number of selected clusters in the objective. We conduct an initial experiment by changing the ILP

---

[6]The reason that we do this is for fair comparison with prior clustering methods.

| reference explanation | proposed explanation | recall |
|---|---|---|
| " *Company*" | " *technology*" | 31 |
| " *Educational Institution*" | " *language*" | 18 |
| " *Artist*" | " *music*" | 38 |
| " *Athlete*" | " *sports*" | 99 |
| " *Office Holder*" | " *politics*" | 79 |
| " *Mean Of Transportation*" | " *technology*" | 63 |
| " *Building*" | " *architecture*" | 43 |
| " *Natural Place*" | " *botany*" | 36 |
| " *Village*" | " *language*" | 67 |
| " *Animal*" | " *zoology*" | 80 |
| " *Plant*" | " *botany*" | 96 |
| " *Album*" | " *music*" | 57 |
| " *Film*" | " *film*" | 97 |
| " *Written Work*" | " *literature*" | 45 |
| **Selector Score** | | 61 |

Table 10: Select-stage evaluation on (DB)pedia of our selection ILP algorithm, the used proposer is `gpt-3.5-turbo`, and the reference assigner is `flan-t5`.

objective to=

$$\mathcal{L} = a \cdot \mathbf{1} + 10 * (s \cdot \mathbf{1}), \qquad (10)$$

and show the results in Table 12. Notably, by not specifying the cluster number to select, our algorithm is able to pick a more compact set of clusters with almost similar coverage.

## G   Further Ablations

We conducted two ablations for PAS to study the contribution of 1) proposing multiple iterations and 2) our selection algorithm. For 1) we compared to only running PAS for 1 iteration, where the proposer "sees" much fewer samples; for 2) we compared to the algorithm of greedily selecting the clusters to maximize coverage and a variant of ILP that sets $\lambda = 0$ (not penalizing the overlaps).

We report the performance in Table 13. Overall, running PAS for five iterations improves over one iteration and using an ILP algorithm with a positive $\lambda$ improves the performance.

## H   PAS-Generated Descriptions

We compare the PAS-generated explanations to the reference explanations; for each pair of generated and reference explanations, we also compute the $F_1$ score between the two generated and the reference cluster (100 if they are identical and 0 if they are disjoint). Generally we found that the generated explanations are semantically relevant or even equivalent to the references (Table 14, 15, 17, 16, 18, and 21); the only exception is when we used `gpt-3.5-turbo` as the proposer and `Flan-T5` as

| Dataset | Propose Score | Select Score | # matched selected proposes |
|---------|--------------|--------------|------------------------------|
| (DB)    | 80           | 62           | 10                           |
| (NYT)   | 86           | 69           | 7                            |

Table 11: Select-stage evaluation on the 14-class (DB)pedia and 9-class (NYT) Topics of our selection ILP algorithm. The score and the # of proposes are averaged over three runs. The number of selected proposes is the same as the class number, as it is enforced in the algorithm.

| Dataset | Method | Select Score | # selected proposes | # matched selected proposes |
|---------|--------|--------------|---------------------|------------------------------|
| (DB)  | ILP w/ # Clusters Constraint | 62 | 14 | 10 |
| (DB)  | ILP w/ # Clusters Penalty    | 59 | 8  | 8  |
| (NYT) | ILP w/ # Clusters Constraint | 69 | 9  | 7  |
| (NYT) | ILP w/ # Clusters Penalty    | 67 | 7  | 7  |

Table 12: Select-stage evaluation on (DB)pedia and (NYT) Topics comparing the ILP method in our paper where we enforce the number of selected clusters with a constraint and a variation where we inject a small cost to favor small number of clusters. The score and the # of proposes are averaged over three runs.

the assigner to cluster based on writing styles on SYNGOALEX (Table 19), but the problem alleviates when we use better models (gpt-4 as proposer, gpt-3.5-turbo as the assigner, Table 20).

# I  OPENGOALEX Datasets

Most of our problems are adapted from the OPEND5 dataset from Zhong et al. (2023). To save budget, for each corpus we randomly sampled 400 text samples.

**human-written-feedback.** human-written feedback for model-generated summaries (Scheurer et al., 2023), with the goal of *"categorizing model errors*.

**abc-headlines**. We collect headlines published by ABC news, an American news company from Kulkarni (2018). ABC headlines are directly downloaded from Harvard Dataverse. The year is extracted from the publication date field. Samples are constructed from the headline text. The goal is to cluster based on the topic of the news. The data is downloadable from https://doi.org/10.7910/DVN/SYBGZL with license CC0 1.0.

**amazon-reviews**. We collect Amazon reviews collected from various product categories from Ni et al. (2019). Amazon reviews are downloaded from a 2018 crawl of the website. The goal is to cluster based on what aspects did the customer complained about the product. The dataset can be downloaded at https://nijianmo.github.io/amazon/index.html. We considered three categories: Beauty product, electronics, and office products.

**rate-my-prof**. We collect reviews of lecturers from RateMyProfessor.com from He (2020). We download a sample of RateMyProfessor.com reviews from an online repo. The goal is to cluster based on what aspects did the students comment on the teacher. This dataset can be downloaded from https://data.mendeley.com/datasets/fvtfjyvw7d/2 under CC BY 4.0 .

**debate-arguments** arguments for a position (Habernal and Gurevych, 2016), with the goal of *"categorizing the types of arguments"*. We took the subset of arguments for the position *"why spanking is bad"*, *"why william farquhar ought not to be honoured as the rightful founder of singapore"*, and *""tv is better than books"*.

**clickbait-headlines** We collect headlines across time from the Examiner, a clickbait news site from Kulkarni (2020). The Examiner headlines are directly downloaded from Kaggle. Samples are constructed from the headline text. The goal is to cluster based on their topics. The dataset can be downloaded at https://www.kaggle.com/datasets/therohk/examine-the-examiner, with license CC0: public domain.

**happy-moments**. We collect self-reported happy moments and demographic characteristics from Asai et al. (2018). The HappyDB dataset is downloaded from the official GitHub repository. Demographic data is cleaned and merged into happy moments. Happy moment descriptions are treated as samples. The goal is to cluster based on whom did the person spend the happy moments with. This dataset can be downloaded at https:

| Macro F$_1$(%) | (AG) Topic | (DB) Topic | (NYT) Topic | (NYT) Location | (SYN) Topic | (SYN) Language | Style |
|---|---|---|---|---|---|---|---|
| PAS | 87 | **71** | 70 | **76** | **98** | 97 | **31** |
| Only 1 Iteration | **88** | 60 | 48 | **77** | 95 | 97 | 29 |
| Selection w/ $\lambda = 0$ | 74 | **71** | 57 | 72 | 98 | **99** | 29 |
| Selection w/ Greedy | 70 | 65 | 55 | 68 | 92 | 98 | 28 |

Table 13: PAS with ablations without the iterative proposing technique and with different selection algorithms described in Section 4.4. Overall, running multiple iterations and using ILP with $\lambda > 0$ are helpful.

| reference explanation | output explanation | F$_1$ |
|---|---|---|
| *"has a topic of Politics"* | *"has a topic of politics and social issues"* | 84 |
| *"has a topic of Sports"* | *"has a topic of sports"* | 97 |
| *"has a topic of Business"* | *"has a topic of finance"* | 81 |
| *"has a topic of Technology"* | *"has a topic of technology"* | 85 |

Table 14: AG's News, clustering based on Topics, proposer=gpt-3.5-turbo, assigner=flan-t5

//github.com/megagonlabs/HappyDB under unknown license.

**yc−startups**. We collect descriptions of companies that were part of the Y Combinator startup incubator from Bhalotia (2022). YCombinator company descriptions are downloaded from a 2022 scrape on GitHub. Only companies with long descriptions are preserved. The goal is to cluster based the type of startups. The dataset can be downloaded from https://www.kaggle.com/datasets/benhamner/y-combinator-companies.

## J  Explainability Evaluation Instance

**Explanation by PAS.** To reduce the workload of the crowdworkers, we only showed them a condensed summary of the explanation. For example, for the explanation:

*"whether this feedback advises adding omitted details; specifically, the feedback points out that certain key details or aspects are missing from the text, which is necessary for a complete understanding. For example, 'The summary should include specific details about why things in their day went wrong."*

we only showed the workers

*"whether this feedback advises adding omitted details"*.

This reduces our average explanation length to be around 5 words.

**Keyword-based Explanations for LDA and Instructor.** As PAS-generated explanations have around 5 terms on average, for each of the LDA and Instructor generated cluster, we choose 5 terms to represent and explain the cluster.

Instructor is only able to produce clusters of text based on representations. We couple it with a representative term mining method that is commonly used in text mining and taxonomy construction (Zhang et al., 2018; Shang et al., 2020; Mekala and Shang, 2020). The method involves first identifying a vocabulary that the representative terms might fall in. This step is usually done by applying AutoPhrase (Shang et al., 2018) on the entire text corpus and thresholding the unigrams and multigrams found. Then, for each cluster, we assign each term in the vocabulary a score based on statistical signals specific to that cluster that correspond to popularity, discriminativeness and informativeness[7]. This score can also be seen as a generalized version of tf-idf. Finally, for each cluster, we take the top 5 terms as its explanation.

LDA is able to word-based explanations for topic clusters by itself. However, we found the word clusters LDA generated lack in quality, despite stop word pruning and tf-idf reweighting. We therefore first apply LDA to obtain the topic clusters, and then use the same representative term mining method above to find the top 5 terms.

**Implementation Details.** For Autophrase, we use the official implementation at https://github.com/shangjingbo1226/AutoPhrase, and do not change the distant supervision or stop words list that was provided. We apply a cutoff threshold of

---

[7]While this score has slightly different definitions in different papers, we follow the one from Shang et al. (2020)

| reference explanation | output explanation | $F_1$ |
|---|---|---|
| *"has a topic of Company"* | *"has a topic of business"* | 75 |
| *"has a topic of Educational Institution"* | *"has a topic of education"* | 91 |
| *"has a topic of Artist"* | *"has a topic of rivers"* | 0 |
| *"has a topic of Athlete"* | *"has a topic of sports and recreation"* | 93 |
| *"has a topic of Office Holder"* | *"has a topic of politics and government"* | 90 |
| *"has a topic of Mean Of Transportation"* | *"has a topic of military equipment"* | 82 |
| *"has a topic of Building"* | *"has a topic of architecture"* | 82 |
| *"has a topic of Natural Place"* | *"has a topic of mountains"* | 56 |
| *"has a topic of Village"* | *"has a topic of villages and towns"* | 99 |
| *"has a topic of Animal"* | *"has a topic of lakes"* | 3 |
| *"has a topic of Plant"* | *"has a topic of biology"* | 73 |
| *"has a topic of Album"* | *"has a topic of music"* | 75 |
| *"has a topic of Film"* | *"has a topic of cinema"* | 89 |
| *"has a topic of Written Work"* | *"has a topic of literature"* | 70 |

Table 15: DBpedia, clustering based on Topics, proposer=`gpt-3.5-turbo`, assigner=`flan-t5`

| reference explanation | output explanation | $F_1$ |
|---|---|---|
| *"has a location of iraq"* | *"has a location of Iraq"* | 63 |
| *"has a location of russia"* | *"has a location of Russia"* | 77 |
| *"has a location of japan"* | *"has a location of japan"* | 86 |
| *"has a location of canada"* | *"has a location of canada"* | 81 |
| *"has a location of britain"* | *"has a location of Britain"* | 87 |
| *"has a location of france"* | *"has a location of France"* | 82 |
| *"has a location of germany"* | *"has a location of germany"* | 79 |
| *"has a location of america"* | *"has a location of the United States"* | 48 |
| *"has a location of china"* | *"has a location of china"* | 84 |
| *"has a location of italy"* | *"has a location of italy"* | 94 |

Table 16: NYT, clustering based on Locations, proposer=`gpt-3.5-turbo`, assigner=`flan-t5`

| reference explanation | output explanation | $F_1$ |
|---|---|---|
| *"has a topic of health"* | *"has a topic of healthcare"* | 78 |
| *"has a topic of estate"* | *"has a topic of housing and living situations"* | 80 |
| *"has a topic of politics"* | *"has a topic of war and weapons"* | 68 |
| *"has a topic of science"* | *"has a topic of climate change"* | 28 |
| *"has a topic of sports"* | *"has a topic of sports and competition"* | 97 |
| *"has a topic of business"* | *"has a topic of business and economics"* | 77 |
| *"has a topic of arts"* | *"has a topic of art exhibition"* | 63 |
| *"has a topic of technology"* | *"has a topic of technology and communication"* | 60 |
| *"has a topic of education"* | *"has a topic of education"* | 82 |

Table 17: NYT, clustering based on Topics, proposer=`gpt-3.5-turbo`, assigner=`flan-t5`

| reference explanation | output explanation | $F_1$ |
|---|---|---|
| *"has a natural language of English"* | *"has a natural language of english"* | 95 |
| *"has a natural language of French"* | *"has a natural language of french"* | 99 |
| *"has a natural language of Deutsch"* | *"has a natural language of german"* | 96 |
| *"has a natural language of Spanish"* | *"has a natural language of spanish"* | 100 |

Table 18: SYNGOALEX, clustering based on Language, proposer=`gpt-3.5-turbo`, assigner=`flan-t5`

| reference explanation | output explanation | $F_1$ |
|---|---|---|
| *"has a writing style of twitter"* | *"has a writing style of health and wellness advice"* | 32 |
| *"has a writing style of screen play"* | *"has a writing style of news article"* | 42 |
| *"has a writing style of rap"* | *"has a writing style of instructional text"* | 22 |
| *"has a writing style of poem"* | *"has a writing style of artistic description"* | 28 |

Table 19: SYNGOALEX, clustering based on Style, proposer=`gpt-3.5-turbo`, assigner=`flan-t5`

| reference explanation | output explanation | $F_1$ |
|---|---|---|
| *"has a writing style of twitter"* | *"has a writing style of instructional or informat..."* | 45 |
| *"has a writing style of screen play"* | *"has a writing style of narrative or storytelling..."* | 51 |
| *"has a writing style of rap"* | *"has a writing style of using rhymes and rhythm"* | 49 |
| *"has a writing style of poem"* | *"has a writing style of incorporating foreign lan..."* | 34 |

Table 20: SYNGOALEX, clustering based on Style, proposer=`gpt-4`, assigner=`gpt-3.5-turbo`

| reference explanation | output explanation | $F_1$ |
|---|---|---|
| *"has a topic of what sports to do to improve your..."* | *"has a topic of sports and physical activity"* | 98 |
| *"has a topic of a new anime has been announced"* | *"has a topic of anime and animation"* | 99 |
| *"has a topic of a tech company releases a new gro..."* | *"has a topic of advanced technology"* | 99 |
| *"has a topic of how to improve your productivity"* | *"has a topic of workplace productivity"* | 97 |

Table 21: SYNGOALEX, clustering based on Topics, proposer=`gpt-3.5-turbo`, assigner=`flan-t5`

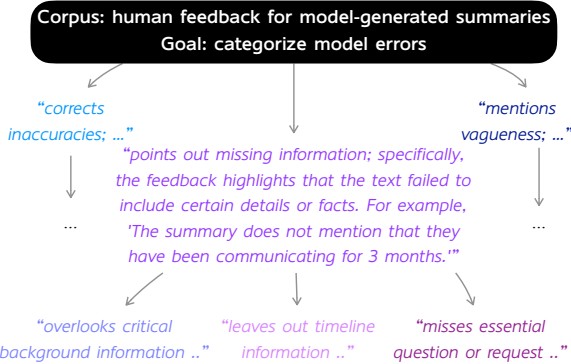

Figure 12: By applying PAS hierarchically to a corpus of human-written comments for model-generated summaries (Scheurer et al., 2023), we can automatically induce taxonomies of error categories for a text summarization system. The texts in quotes are copies (sometimes abbreviated) from PAS's output.

0.5 for unigram and 0.8 for multigram which we tested to work well on heldout data.

For representative term mining, we re-implement the representativeness score from the Shang et al. (2020) by ourself.

For LDA, we use the implementation in `sklearn`.

**Sanity Check on Keyword Based Implementation.** To ensure that our implementation of keyword-based explanation is reasonable, we applied LDA and Instructor to cluster topics on the English subset of SYNGOALEX, a task that we know that they can achieve better performance. As expected, Instructor achieves 80% accuracy while LDA achieves 76%, which is much higher than the performance on OPENGOALEX.

**HIT Task.** We paid crowdworkers $0.05 for each binary choice of explanations. The authors on average can perform 4 HITs per minute, which translates to around $12/hour of payment. We recruited Turkers with > 98% of HIT acceptance rate in the history.

## K   Additional Example Taxonomy

We provide additional example taxonomy over model errors and customer reviews in Figure 12 and 13.

## L   Implementation Details

In terms of software libraries, we used `pulp` (Mitchell et al., 2011) to implement ILP for PAS; we used `transformers` (Wolf et al., 2020) to run `flan-t5-xl`, `e5-large`, and `instructor-xl`; we used `sklearn` (Pedregosa et al., 2011) to run LDA and K-means on text embeddings.

## M   Sensitivity Study

**Prompt Sensitivity Study.** We conduct a study to understand how sensitive our method is to our crafted prompts. We perturb our proposer and assigner prompt template by paraphrasing it extensively; the perturbed prompts are in Appendix A. The results are shown in Table 22. The performance does not change much when the prompt changes. Most importantly, the two claims, (1) our method is on par with prior clustering methods on topic clustering, and (2) our method is much better than prior methods when the goal is non-topic, still holds robustly.

**Dataset Sensitivity Study.** Real world data is usually not clean and perfectly balanced. To understand how the data balance and data noise affects our method, we conduct the following two studies on the (DB)pedia dataset. For the imbalanced scenario, we randomly sampled 7 classes and removed half of their data points. Then among the 7 classes, we further random sampled 3 classes and again removed half of their remaining data points. In Table 23, we report the delta changes of Instructor and our method PAS, and observe that the change is similar when the dataset becomes imbalanced.

For the noisy experiment, we randomly sampled 4 classes and removed $\frac{7}{8}$ of their data points. Then, we consider the clustering problem on the 10 remaining classes, but with the additional data of the 4 removed classes as extra noise. The evaluation is only done on the data points of the 10 remaining classes, but the model needs to be robust to noisy data points. From Table 23, we observe that both Instructor and our method have a similar small drop in performance.

Both results indicate that our method is not especially vulnerable to imbalance or noise in the dataset.

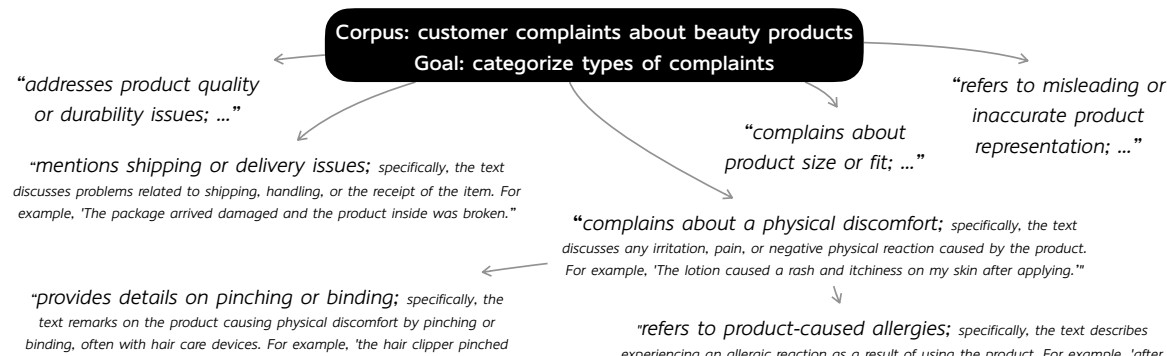

Figure 13: Example taxonomy for complaints about beauty products from Amazon (He and McAuley, 2016).

| Macro F$_1$ (%) Goal | (AG) | (DB) | (NYT) | (SYN) | (NYT) | (SYN) | (SYN) |
|---|---|---|---|---|---|---|---|
| | | | Topic | | Location | Language | Style |
| Instructor | 84 | 82 | 69 | 77 | 54 | 25 | 25 |
| PAS | 87 | 71 | 70 | 98 | 76 | 97 | 31 |
| PAS (new proposer prompt) | 86 | 72 | 68 | 98 | 75 | 98 | 28 |
| PAS (new assigner prompt) | 87 | 63 | 67 | 98 | 82 | 98 | 29 |

Table 22: We paraphrase the prompt used for the proposer and assigner and assess their performance.

| Delta Macro F$_1$ (%) | sample seed | | |
|---|---|---|---|
| | seed = 0 | seed = 1 | seed = 2 |
| *Imbalance* | | | |
| Instructor | -11 | -14 | -10 |
| PAS | -10 | -8 | -12 |
| *Noise* | | | |
| Instructor | -1 | -1 | -2 |
| PAS | -2 | -3 | -3 |

Table 23: We create three imbalanced and noisy versions of the DBpedia dataset where the difference is at the random seed during data creation. We calculate the difference between the performance on the clean dataset of Instructor and PAS.