# OpenReview forum: "Goal-Driven Explainable Clustering via Language Descriptions"
_EMNLP/2023/Conference — EMNLP 2023 Main_

### Official Review · Reviewer_wCff · 2023-08-04

**Typos Grammar Style And Presentation Improvements:** 648
**Soundness:** 4

**Excitement:**

4: Strong: This paper deepens the understanding of some phenomenon or lowers the barriers to an existing research direction.

**Paper Topic And Main Contributions:**

This paper proposes **a new task** called "Goal-Driven Clustering with Explanations" (GOALEX) considering the users' goals and produces an explainations of a  clusters' meanings. The input to GOALEX is a corpus of texts with multiple attributes and a goal description in natural language (e.g., "I want to cluster based on sentiment"), and the output is a set of clusters, each with a natural language explanation of which text samples should or should not belong to the cluster. The paper also presents a **three-stage algorithm** called Propose-Assign-Select (PAS) to tackle GOALEX: (a) prompting a language model to generate a list of goal-related explanations for candidate clusters, (b) assign text samples only to the explanations that support them, and (c) uses integer linear programming to search for a subset of candidate explanations so that each sample is roughly supported once. PAS was evaluated on known and open-ended corpora and was found to produce more accurate and goal-related explanations than prior methods (e.g., LDA).

**Reasons To Accept:**

- proposing a task that has practical application for data analysis and solves the hard problem of determining meaning fully labels for clusters
- Well written paper
- thorough analysis and experiments


**Reasons To Reject:**

- the solution with the best performance depends on closed LLMs (GPT-4/claude-v1.3)
  - high cost
  - because training data for these LLMs hasn't been released, there could be test data leakage
[the authors address the potential for leakage in appendix E and I mostly agree with their reasoning]
- the authors refer to discussions/tables in the appendix in the main text (e.g., 277, 384).
- difficult to reproduce because the work relies on closed LLMs that (a) are non-deterministic in their output and (b) can change at any time

**Reproducibility:**

3: Could reproduce the results with some difficulty. The settings of parameters are underspecified or subjectively determined; the training/evaluation data are not widely available.

**Reviewer Confidence:**

3: Pretty sure, but there's a chance I missed something. Although I have a good feel for this area in general, I did not carefully check the paper's details, e.g., the math, experimental design, or novelty.

---

> ### Author Rebuttal · Authors · 2023-08-29
>
> > Q: Reproducibility of the results.
>
> While the exact same performance reported in our paper might not be deterministically replicable due to API changes, the main claims of our paper are robustly reproducible. To illustrate this, we conducted an experiment with gpt-3.5-turbo-0301 and gpt-3.5-turbo-0613, two different gpt 3.5 models and show their results in the table below.
>
> |                          | AGNews | DBpedia | NYT Topic | SYN Topic | NYT Location | SYN Language | SYN Style |
> |--------------------------|--------|---------|-----------|-----------|--------------|--------------|-----------|
> | Instructor               | 84     | 82      | 69        | 77        | 54           | 25           | 25        |
> | PAS (paper)              | 87     | 71      | 70        | 98        | 76           | 97           | 31        |
> | PAS (gpt-3.5-turbo-0301) | 87     | 69      | 67        | 98        | 81           | 99           | 29        |
> | PAS (gpt-3.5-turbo-0613) | 88     | 68      | 64        | 98        | 75           | 98           | 28        |
>
> The main claims from our paper still robustly hold even under different API endpoints.
> - Our method is on par with prior clustering methods on topic clustering. By looking at the columns with AGNews, DBpedia, NYT Topic and SYN Topic this claim is still supported regardless of the differences in the LLM. As already acknowledged in our submission, our method has lower performance on DBpedia due to the cluster merging effect illustrated in our paper (paragraph L356-368).
> - Our method is much better than prior methods when the goal is non-topic. This is also supported with the numbers in columns NYT Location, SYN Language and SYN Style.
>
> > Q: Method requires high-end LLMs which may be of high costs
>
> We acknowledged this in our limitations section, and we are working on a follow-up project of this paper to reduce this cost.
>
> We’d also like to point out that when we set our proposer as gpt3.5, and validator as t5-xl (the majority of our experiments in section 4), the cost is affordable. One would need about **0.1$** in total for querying the openly accessible api for gpt 3.5, and **a single 24G gpu** to run inference with t5-xl.
>
> We agree that the current best performance relies on the most powerful large models but we hope that the cost would decrease soon due to algorithmic and hardware improvements.
>
> > Q: Referring to discussions/tables in the appendix.
>
> Thanks for your recommendation. We believe that the content in the main paper is self-contained and does not rely on the appendix. However, we agree that it would be better to move some discussion to the main paper when an additional page is available. Concretely, we will move L277 (validity of using LLMs), L384 (syn style with gpt 4 as proposer and gpt 3.5 as assigner) to the main text as suggested. Further, we will move Appendix L (experiments on an additional clustering method E5) to the main text as well in the updated version. Please feel free to point out more places where the presentation could be improved.

---

### Official Review · Reviewer_v3tK · 2023-08-06

**Soundness:** 3

**Excitement:**

4: Strong: This paper deepens the understanding of some phenomenon or lowers the barriers to an existing research direction.

**Paper Topic And Main Contributions:**

This paper aims to make unsupervised text clustering more user-centric and explainable. It proposes a text clustering task that considers the users' goals and explains the meanings of detected clusters as free-form language descriptions. A three-stage algorithm, Propose-Assign-Select (PAS), is proposed to tackle the task. The algorithm prompts a language model to generate goal-related explanations for candidate clusters, assigns text samples to the explanations that support them, and then uses integer linear programming to select a subset of candidate clusters that cover most samples while minimizing overlaps. The authors also evaluate the PAS algorithm under both automatic and human evaluation on corpora with or without labels

**Reasons To Accept:**

- The paper introduces a practical method for making text clustering understandable, useful across various real-world applications.
- Implementation details are provided for replicating the results.
- The paper is easy to read with intuitive figures.

**Reasons To Reject:**

- How much does the output vary with tiny variations of prompts/instructions to the models? Any sort of sensitivity analysis may help measure the robustness.
- It would also make the experiment more solid if the author could explain how the proposed method handles clusters of different sizes, noise in the data, and high-dimensional data.

**Reproducibility:**

5: Could easily reproduce the results.

**Reviewer Confidence:**

2: Willing to defend my evaluation, but it is fairly likely that I missed some details, didn't understand some central points, or can't be sure about the novelty of the work.

---

> ### Author Rebuttal · Authors · 2023-08-29
>
> Thanks for appreciating our task, method and writing! Please see our response below.
>
> > Q: Is our method sensitive to different prompt formats?
>
> Following your recommendation, we **include a new experiment** (details below) to study the sensitivity of our results to different proposer and assigner prompts, and we find that 1) the change in performance is minor in most datasets, and 2) the conclusions in our paper still robustly hold. Thanks for your insightful comments which make our paper stronger!
>
> Details for the experiments:  We analyze the sensitivity of the proposer and the assigner to the prompt format; we paraphrase the prompt extensively while not changing its meaning (prompt attached at the end). The table below shows the results
>
> |                           | AGNews | DBpedia | NYT Topic | SYN Topic | NYT Location | SYN Language | SYN Style |
> |---------------------------|--------|---------|------------|-----------|--------------|--------------|-----------|
> | Instructor                | 84     | 82      | 69         | 77        | 54           | 25           | 25        |
> | PAS (default)             | 87     | 71      | 70         | 98        | 76           | 97           | 31        |
> | PAS (new proposer prompt) | 86     | 72      | 68         | 98        | 75           | 98           | 28        |
> | PAS (new assigner prompt) | 87     | 63      | 67         | 98        | 82           | 98           | 29        |
>
> The performance does not change much when the prompt changes. Most importantly, the following major claims in our paper still hold robustly:
> - Our method is on par with prior clustering methods on topic clustering. Our method is indeed lacking on DBpedia, due to a cluster merging effect we illustrated in the paper (paragraph L356-368)
> - Our method is much better than prior methods when the goal is non-topic.
>
> > Q: Additional experiment of clusters of different sizes, noise in the data, and high-dimensional data.
>
> We conducted the following two preliminary studies regarding data size and noise on the DBpedia dataset (14 classes). We found that **our method is as insensitive as our baseline method, Instructor, to imbalanced or noisy data.**
>
> For clustering on imbalanced data, we randomly sampled 7 classes and removed half of their data points. Then among the 7 classes, we further random sampled 3 classes and again removed half of their remaining data points. We repeat this procedure 3 times and create 3 imbalanced datasets by varying the sampled classes. In the table below, we report the delta changes of Instructor and our method PAS, and observe that the change is similar when the dataset becomes imbalanced. This shows that our method is not more vulnerable to imbalanced data.
>
> | DBpedia Imbalance (delta) | sample seed = 0 | sample seed = 1 | sample seed = 2 |
> |---------------------------|-----------------|-----------------|-----------------|
> | Instructor                | -11             | -14             | -10             |
> | PAS                       | -10             | -8              | -12             |
>
> For the noisy experiment, we randomly sampled 4 classes and removed 7 / 8 of their data points. Then, we consider the clustering problem on the 10 remaining classes, but with the additional data of the 4 removed classes as noise. The evaluation is only done on the data points of the 10 remaining classes, but the model needs to be robust to the data points not belonging to any of them. We observe that both Instructor and our method have a similar small drop in performance, showing that our method is also not especially sensitive to noise in the dataset.
>
> | DBpedia Noise (delta) | sample seed = 0 | sample seed = 1 | sample seed = 2 |
> |-----------------------|-----------------|-----------------|-----------------|
> | Instructor            | -1              | -1              | -2              |
> | PAS                   | -2              | -3              | -3              |
>
> Can you clarify what “high-dimensional data” refers to?
>
> > Here are the original and changed proposer and assigner prompts.
>
> Original Proposer Prompt:
> ```
> In this task you will need to come up with categories to group the text together based on a goal, where each category can be described with a natural language predicate. For example
>
> 0. How are you doing?
> 1. How do I apply for visa?
> 2. Isn't it ridiculous?
> 3. How to get there by bus?
> 4. How is it going?
> 5. Aren't technological advancement good for the humanity?
>
> Here is our goal: I want to cluster the questions based on their functionalities. Your responses are:
> - "is a rhetorical quesiton"
> - "is meant for greetings"
> - "is asking for practically useful information"
>
> (note that all of them need to be natural language predicates that can be validated on the given samples)
>
> Here are some texts:
>
> {samples_in_prompt}
>
> Here is our goal: {goal}
> Please suggest me a few descriptions, one in a line, starting with "-" and surrounded by quotes"". In terms of formatting, each of them needs to be a predicate about a text, for example:
> - "uses double negation"
> - "has a conservative stance"
>
> Do not output anything else. (Note that the examples might not be goal related, and your response should be both formatted correct as above and related to the goal.)
> {example_description_in_prompt}
>
> Again, here's the goal: {goal}. Your responses are:
> - "
> ```
>
> New Proposer prompt:
> ```
> In the given task, you are required to devise categories to classify the texts according to a specific objective. Each category should be expressible via a natural language predicate. Here's an example:
>
> 0. How are you doing?
> 1. How do I apply for a visa?
> 2. Isn't it ridiculous?
> 3. How to get there by bus?
> 4. How is it going?
> 5. Aren't technological advancements good for humanity?
>
> The objective here is: I am looking to categorize these questions based on their purpose. Your response could be:
> - "is a rhetorical question"
> - "is intended for pleasantries"
> - "is asking for practical information"
>
> (remember these responses must be written as natural language predicates which can be verified against the provided sample texts)
>
> Here are some text samples:
>
> {samples_in_prompt}
>
> The objective is: {goal}
> Please can you provide a number of descriptors, beginning with "-" and are within "". For formatting purposes, every response must be a predicate about a text, for instance:
> - "employs double negation"
> - "exhibits a traditional standpoint"
>
> Do not provide any other outputs. (Note that these examples might not correspond to the actual objective, yet your response should both adhere to the above-mentioned formatting and be pertinent to the stated goal.)
> {example_description_in_prompt}
>
> Once more, the objective is: {goal}. Your responses could be:
> - "
> ```
>
> Original Assigner Prompt:
> ```
> Check whether the TEXT satisfies a PROPERTY. Respond with Yes or No. When uncertain, output No.
>
> Now complete the following example -
> input: PROPERTY: {candidate_explanation}
> TEXT: {text}
> output:
> ```
>
> New Assigner Prompt:
>
> ```
> Determine if the TEXT adheres to a certain PROPERTY. Give a Yes or No response. When in doubt, lean towards No.
>
> Now, proceed with the following example -
> input: PROPERTY: {candidate_explanation}
> TEXT: {text}
> output:
> ```

---

### Official Review · Reviewer_8dya · 2023-08-07

**Soundness:** 3

**Excitement:**

3: Ambivalent: It has merits (e.g., it reports state-of-the-art results, the idea is nice), but there are key weaknesses (e.g., it describes incremental work), and it can significantly benefit from another round of revision. However, I won't object to accepting it if my co-reviewers champion it.

**Paper Topic And Main Contributions:**

This paper introduced a novel text clustering formulation, GOALEX, which considers users' goals and output natural language explanations for clusters. The author proposed a multi-stage algorithm, PAS, to address this new problem. The automatic/human-involved evaluation results show that the proposed method produces more accurate and goal-oriented clusters compared to existing methods.

**Reasons To Accept:**

1. I personally think the proposed formulation has practical value, so the tasks and methods based on it are application-oriented.
2. The proposed method (PAS) is intuitive, effectively solves GOALEX based on the experimental results.
3. The paper presents new formulation, dataset, and detailed methodology using LLMs. This provides value and insights to the research community.

**Reasons To Reject:**

1. Potential unfair comparisons. The authors use LDA and Instructor as baselines, however, LDA is a classic work from 20 years ago, and Intructor-XL only has 1.5B parameters, which are far from the proposed PAS using gpt-3.5-turbo.
2. The writing clearly describes the approach but does not sufficiently highlight the contributions. More emphasis on the key innovations would improve readability.
3. There are some typos, formatting errors, and incomplete paragraphs needing further proofreading (Please refer to typo field).

**Reproducibility:**

4: Could mostly reproduce the results, but there may be some variation because of sample variance or minor variations in their interpretation of the protocol or method.

**Reviewer Confidence:**

3: Pretty sure, but there's a chance I missed something. Although I have a good feel for this area in general, I did not carefully check the paper's details, e.g., the math, experimental design, or novelty.

**Typos Grammar Style And Presentation Improvements:**

- Table 4: Missing top line.
- Line 648 Incorrect hyphen.
- Line 647 Incomplete paragraph.
- Line 1114 Redundant "=" symbol.

Additionally, reducing the underlining in Sections 4 and 5 may improve the readability, as they do not indicate core ideas. Same as the bracketed dataset abbreviations in Section 4.

---

> ### Author Rebuttal · Authors · 2023-08-29
>
> Thanks for appreciating our method and task formulation. Please see our response below.
>
> > Q: Is the comparison fair if the baselines use smaller models than LLMs?
>
> We did not try to claim that our method is superior to previous methods developed for topic clustering. This is not an apple-to-apple comparison in the first place, since our task formulation is different: prior methods could neither condition on specific goals nor provide natural language explanations. We ran our method on existing benchmarks (e.g., Table 1) only to help the readers develop a calibrated and contextualized understanding of what our system can achieve.
>
> To best help the readers contextualize the results, we tried the best-performing embedding-based clustering systems available to us. We took Instructor (and also another method E5 in the Appendix L) as the major reference point for performance since (1) it is the state of the art clustering method and (2) it was claimed to be able to follow instructions given to the embedder, and (3) it has the same size of our validator (1.5B parameter encoder). However, even though we tried various prompting strategies and reported the best results (Section 4.4), it still fails to condition on the goal.
>
> Indeed, it might have been more informative to include results that cluster based on GPT-3.5 embeddings; however, 1) they are not accessible, and 2) task formulation-wise, pure embedding based methods cannot condition on goals specified in natural language or provide explanations for each cluster.
>
> > Q: The innovation of this paper has not been sufficiently highlighted.
>
> We summarized our contribution (a new task formulation, a new method, and example applications) in the conclusion section, but we do agree that the contribution could have been emphasized more. In the updated version, we will also summarize the contribution at the end of the introduction.
>
> > Typos and other stylistic comments
>
> Thanks for reading our paper carefully and we will fix all of them in our updated version. Regarding the underline and parentheses around method and dataset names – we added them because some other readers complained that there are too many entities in the paper, and adding these notations helped them distinguish the type of a noun (e.g., method or dataset).

---

### Official Review · Reviewer_qK88 · 2023-08-11

**Typos Grammar Style And Presentation Improvements:** Nothing was found to the best of my u…
**Soundness:** 4

**Excitement:**

4: Strong: This paper deepens the understanding of some phenomenon or lowers the barriers to an existing research direction.

**Missing References:**

No missing reference.

**Paper Topic And Main Contributions:**

This paper proposes an approach for personalized and goal-oriented clustering with explanations for each cluster. Given a corpus, string describing the goal of the clustering process and the number of clusters, the approach GOALEX outputs the number of clusters each accompanied by an explanation. To do this, the authors rely on the algorithm PAS which can generate goal-related explanations for
the clusters. The method is based on prompting a language model to obtain a list of goal-related explanations, then assigning each explanation to a cluster and finally selecting the relevant cluster by maximizing the coverage while minimizing the overlap. The authors propose two different evaluation approaches: one based on automatic evaluation by comparing their proposed method to sota methods approaches on clustering datasets, and the second method is based on manual evaluation on open-end corpora.

**Questions For The Authors:**

see reasons to reject.

**Reasons To Accept:**

- The task of obtaining personalized and goal-oriented clusters with explanations describing the content is interesting.
- Workflow figure (Figure 2) is clear and understandable.
- The paper is clearly written and easy to understand.
- Interesting approach and creative applications to generating taxonomies.

**Reasons To Reject:**

- to the best of my understanding, in the main paper, it is assumed that the list of explanations given by the LM is correct without any way of evaluating if this assumption holds and how the result changes based on this fact.

**Reproducibility:**

4: Could mostly reproduce the results, but there may be some variation because of sample variance or minor variations in their interpretation of the protocol or method.

**Reviewer Confidence:**

3: Pretty sure, but there's a chance I missed something. Although I have a good feel for this area in general, I did not carefully check the paper's details, e.g., the math, experimental design, or novelty.

---

> ### Author Rebuttal · Authors · 2023-08-29
>
> Thanks for appreciating our writing and the new task that we proposed!
>
> > Q: Do LLM-proposed explanations need to be “correct” for our method to work?
>
> Empirically we found that LLM usually produces explanations closely related to the goal and the data. The biggest failure modes is that some of their explanations can be (a) too specific and cover too few data points, (b) too broad and cover too many data points, or (c) too similar to other explanations. **Our algorithm can address all of these failure modes**, as it attempts to select a subset of explanations to cover each data point exactly once: too specific explanations will be penalized because they reduce coverage, while too broad or similar explanations will be penalized because they induce overlaps.  For example, in the nyt-topic dataset, the proposer generated several fine-grained explanations about politics,
> - has a topic of cyberterrorism concerns in Japan
> - has a topic of international relations and diplomacy
> - has a topic of politics
> - has a topic of non-entertainment
>
> Our selection algorithm can correctly identify a single explanation “has a topic of politics” among these, because all of the rest are either too broad or too specific to make our final result cover each sample once.
>
> Additionally, we did inspect the quality of the final selected cluster explanation. In Section 5, we measured the correctness of the final selected explanations based on human evaluation; in Appendix H table 13-20, we listed the final selected explanations for the automatic evaluations for human inspection. Both experiments reveal that the final selected explanations are of high quality.

---

### Meta-Review · Area_Chair_SQJo · 2023-09-18

**Recommendation:** 5

**Metareview:**

This paper presents a new framework for unsupervised clustering of documents, which tries to satisfy the desiderata of goal driven clustering with free-form textual explanations of clusters. Their approach involves prompting a language model to generate candidate explanations, assigning documents to those explanations, and aligning possible matches using an ILP. Using both automatic and human evaluations, the authors find they are able to generate clusters and explanations that conform to the user-driven goals.

Reviewers were intrigued by this task, and found the paper to be clearly written and easy to follow. They also thought this was creative work with practical value in applications.

The reasons to reject given by the reviewers were relatively minor, including pointing out the high computational cost of this method, suggesting additional baselines, and requesting sensitivity analyses. The most serious concern seems to be that the authors to some extent assume that the generated explanations are correct, without adequately verifying this, but the reviewer who pointed this out was still quite favorable about the paper.

Although the reviewers were evenly split on soundness (between Good and Strong), 3 of the 4 reviewers rated their excitement as Strong, suggesting that this would be a reasonable choice for a main conference paper.

---

### Decision · Program_Chairs · 2023-10-07

**Decision:**

Accept-Main

**Comment:**

This paper presents a new framework for unsupervised clustering of documents, which tries to satisfy the desiderata of goal driven clustering with free-form textual explanations of clusters. Their approach involves prompting a language model to generate candidate explanations, assigning documents to those explanations, and aligning possible matches using an ILP. Using both automatic and human evaluations, the authors find they are able to generate clusters and explanations that conform to the user-driven goals.

Reviewers were intrigued by this task, and found the paper to be clearly written and easy to follow. They also thought this was creative work with practical value in applications.

The reasons to reject given by the reviewers were relatively minor, including pointing out the high computational cost of this method, suggesting additional baselines, and requesting sensitivity analyses. The most serious concern seems to be that the authors to some extent assume that the generated explanations are correct, without adequately verifying this, but the reviewer who pointed this out was still quite favorable about the paper.

Although the reviewers were evenly split on soundness (between Good and Strong), 3 of the 4 reviewers rated their excitement as Strong, suggesting that this would be a reasonable choice for a main conference paper.